# Integrating Transcriptomics, Proteomics, and Metabolomics to Investigate the Mechanism of Fetal Placental Overgrowth in Somatic Cell Nuclear Transfer Cattle

**DOI:** 10.3390/ijms25179388

**Published:** 2024-08-29

**Authors:** Xiaoyu Zhao, Shanshan Wu, Yuan Yun, Zhiwen Du, Shuqin Liu, Chunjie Bo, Yuxin Gao, Lei Yang, Lishuang Song, Chunling Bai, Guanghua Su, Guangpeng Li

**Affiliations:** 1State Key Laboratory of Reproductive Regulation and Breeding of Grassland Livestock (R2BGL), Inner Mongolia University, 24 Zhaojun Rd., Hohhot 010070, China; zhaoxiaoyu3233@163.com (X.Z.); wss5330402@126.com (S.W.); souw261146@163.com (Y.Y.); dzw15133339850@163.com (Z.D.); lsq32208183lsq@163.com (S.L.); shengwubcj@163.com (C.B.); m18535078392@163.com (Y.G.); mrknowall@126.com (L.Y.); xiaoshuang2000@126.com (L.S.); chunling1980_0@163.com (C.B.); 2College of Life Sciences, Inner Mongolia University, 24 Zhaojun Rd., Hohhot 010070, China; 3Agricultural Genomics Institute at Shenzhen, Chinese Academy of Agricultural Sciences, Shenzhen 518000, China

**Keywords:** SCNT, cattle, placenta, transcriptomics, proteomics, metabolomics

## Abstract

A major factor limiting the development of somatic cell nuclear transfer (SCNT) technology is the low success rate of pregnancy, mainly due to placental abnormalities disrupting the maternal-fetal balance during pregnancy. Although there has been some progress in research on the abnormal enlargement of cloned bovine placenta, there are still few reports on the direct regulatory mechanisms of enlarged cloned bovine placenta tissue. In this study, we conducted sequencing and analysis of transcriptomics, proteomics, and metabolomics of placental tissues from SCNT cattle (*n* = 3) and control (CON) cattle (*n* = 3). The omics analysis results indicate abnormalities in biological functions such as protein digestion and absorption, glycolysis/gluconeogenesis, the regulation of lipid breakdown, as well as glycerolipid metabolism, and arginine and proline metabolism in the placenta of SCNT cattle. Integrating these analyses highlights critical metabolic pathways affecting SCNT cattle placenta, including choline metabolism and unsaturated fatty acid biosynthesis. These findings suggest that aberrant expressions of genes, proteins, and metabolites in SCNT placentas affect key pathways in protein digestion, growth hormone function, and energy metabolism. Our results suggest that abnormal protein synthesis, growth hormone function, and energy metabolism in SCNT bovine placental tissues contribute to placental hypertrophy. These findings offer valuable insights for further investigation into the mechanisms underlying SCNT bovine placental abnormalities.

## 1. Introduction

Somatic cell nuclear transfer (SCNT) is an asexual reproduction technique in which the nucleus of a somatic cell is transplanted into a denucleated oocyte to create a reconstructed embryo [1]. SCNT technology has important potential applications in agricultural breeding, genetically modified animal breeding, endangered species protection, organ transplantation, and other fields [2,3,4]. Although the application of SCNT technology has successfully produced a variety of mammals, such as sheep [5], mice [6], rats [7], cows [8], pigs [9], cats [10], dogs [11], and macaque monkeys [12], the success rate remains low, with less than 5% of SCNT animals being born and surviving normally. The development of cloned animals is often accompanied by abnormalities, including obesity, immunodeficiency, respiratory diseases, and premature death. The extra-embryonic placenta of cloned animals also showed various abnormalities, such as hypertrophy, the proliferation of trophoblastic binuclear cells, and vascular system dysplasia, which hinder the growth and development of cloned animals [13,14]. It is widely believed that incomplete epigenetic reprogramming in the nuclei of donors is the main cause of the abnormal development of cloned offspring [13]. Consequently, research aimed at improving pregnancy outcomes in SCNT animals has largely focused on correcting abnormal nuclear reprogramming during the SCNT process [15,16]. As a crucial organ during pregnancy, the placenta plays a key role in maintaining the normal growth and development of the fetus through the functions of substance exchange, defense, synthesis, and immunity [17]. Abnormalities in the morphology and structure of the placenta are the main cause of pregnancy failure and the occurrence of various fetal diseases [18]. Although there has been some progress in research on the abnormal enlargement of cloned bovine placenta, there are still few reports on the direct regulatory mechanisms of enlarged cloned bovine placenta tissue. In recent years, with the development of omics technologies such as transcriptomics, proteomics, and metabolomics, a new approach has been provided to fully understand the specific mechanism of biological phenomena. These techniques enable the identification of gene expression in samples, the study of proteins and their complexes, the identification of metabolites, and the interpretation of changes in metabolites under different conditions [19,20,21]. Although omics techniques have been used to explore the mechanisms of placental abnormalities in SCNT animals during pregnancy, a more comprehensive and systematic multi-omics analysis of SCNT bovine placentas remains lacking.

Based on the above background, the primary objective of this study was to elucidate the differences in gene expression, protein expression, and metabolite expression between SCNT and CON bovine placentas. We obtained the transcriptomics, proteomics, and metabolomics profiles of SCNT and CON bovine placental tissues, aiming to reveal the expression differences in multi-omics and their underlying regulatory mechanisms in cloned animals. This analysis of transcriptomic, proteomic, and metabolomic data from SCNT bovine placentas provides a significant theoretical foundation for further exploration into the mechanisms underlying the abnormal development of SCNT bovine placenta during pregnancy.

## 2. Results

### 2.1. RNA Sequencing and Identification of Differential Expression Genes (DEGs)

In our previous study, we observed a high abortion rate in SCNT cattle during the middle-to-late gestation period, which was associated with placental abnormalities, typically manifested as umbilical cord enlargement, placental hypertrophy, and abnormal cotyledon size [22]. To explore the mechanism behind abnormal placental hypertrophy in SCNT cattle, we collected placental tissues from SCNT and CON cattle during late gestation for transcriptome sequencing, focusing on differentially expressed genes’ analysis. The numbers of raw reads were 135,540,352 for SCNT and 138,849,900 for CON. After quality control and filtering, we obtained 133,329,260 and 137,015,210 high-quality, valid reads for SCNT and CON, respectively, with the percentage of Q30 bases being 94.45% and above (Table 1). The mapping efficiencies to the bovine reference genome (ARS-UCD1.2) ranged from 95.83% to 96.48% (Table 1). Principal component analysis (PCA) analysis showed distinct clustering within SCNT and CON bovine placenta groups, while samples between groups were dispersed, indicating consistent replication within groups and significant differences between them (Figure 1a). Correlation analysis confirmed a high degree of similarity between replicate samples from SCNT and CON bovine placental tissues (Figure 1b). Compared to CON bovine placentas, 4985 DEGs were identified in the SCNT placentas, with 2684 genes up-regulated and 2337 genes down-regulated (*p*-Value < 0.05 and |Log2 FC| ≥ 1) (Figure 1c,d). The top 10 DEGs are shown in Table 2. Significantly enriched among these were the following: the trophoblast kunitz domain protein 1 (*TKDP1*) and *TKDP4*, regulating placental trophoblast development; the marker of proliferation ki-67 (*MKI67*) and TIMP metallopeptidase inhibitor 4 (*TIMP4*), involved in hormone regulation and endometrial remodeling; and inosine 5′-monophosphate dehydrogenase 1 *(IMPDH1*), a rate-limiting enzyme in de novo GMP biosynthesis. These findings suggest that trophoblast cell proliferation may be abnormal in SCNT bovine placental tissues, potentially affecting trophoblast cell development and leading to hormone-secretion dysfunction.

### 2.2. DEGs Function Enrichment Analysis

To gain deeper insights into the function of DEGs, we categorized them into up-regulated and down-regulated genes, which were then analyzed through the gene ontology (GO) function and *Kyoto Encyclopedia of Genes and Genomes* (KEGG) pathway enrichment, respectively. The GO enrichment analysis indicated that both up-regulated and down-regulated DEGs were primarily involved in biological processes related to metabolism, including the positive regulation of the macromolecule metabolic process, the cellular metabolic process, the primary metabolic process, and the metabolic process itself (Figure 2a). The top 10 GO functional pathways enriched among up-regulated and down-regulated genes are detailed in Appendix A, respectively. The KEGG enrichment analysis for both up-regulated and down-regulated genes revealed that up-regulated DEGs predominantly participated in signaling pathways related to protein digestion and absorption, ECM-receptor interaction, focal adhesion, and glycolysis/gluconeogenesis. In contrast, down-regulated DEGs were chiefly enriched in pathways involving apoptosis, the regulation of lipolysis in adipocytes, and growth hormone synthesis, secretion, and action (Figure 2b,c). The top 10 KEGG signaling pathways enriched for up-regulated and down-regulated DEGs are presented in Appendix A. These findings from GO and KEGG enrichment analyses suggest that the abnormally expressed genes in the SCNT bovine placenta are significantly involved in processes related to glycolysis, growth hormone synthesis and secretion, and cellular metabolism.

### 2.3. Proteomics Detection and Identification of Different Expression of Proteins (DEPs)

To further investigate the mechanism underlying abnormalities in SCNT bovine placental tissues, we conducted the proteomes of the placental tissues from SCNT and CON cattle. The PCA results showed that the samples were clustered within the group and dispersed between the groups, indicating a significant difference between the samples (Figure 3a). Similarly, correlation analysis demonstrated a high degree of similarity in protein composition within the SCNT and CON samples, with marked differences observed between these groups (Figure 3b). In screening for DEPs between SCNT and CON placental tissues, we identified a total of 823 DEPs in the SCNT placenta relative to CON, among which 399 proteins were up-regulated and 424 proteins were down-regulated (|log2FC| ≥ 0.585 and *p*-Value < 0.05) (Figure 3c,d). The top 10 DEPs are listed in Table 3. Notably, proteins such as the following had the most significant differential expression: colony-stimulating factor 1 (CSF1), which is involved in regulating placental development; PDZ and LIM domain protein 7 (PDLIM7), related to embryonic bone formation; integrin alpha 2 (ITGA2) and integrin beta 4 (ITGB4), which are involved in regulatory gene expression and cell growth; and GTPase activating protein Rab5 interacting protein 1 (GARRE1) and the NADH dehydrogenase subunit 3 (NAD3) protein, involved in energy metabolism.

### 2.4. Functional Enrichment Analysis of Differentially Expressed Proteins

To elucidate the functions of DEPs, we performed enrichment analyses of GO functional pathways and KEGG signaling pathways for selected up-regulated and down-regulated DEPs. GO enrichment analysis indicated that up-regulated DEPs were mainly associated with functions of the plasma membrane, an components of the membrane. In contrast, down-regulated DEPs were primarily involved in the small-molecule metabolic process and the lipid metabolic process, among others (Figure 4a,b). The top ten GO functional pathways enriched via up-regulated and down-regulated DEPs are detailed in Appendix A, respectively. KEGG pathway enrichment analysis revealed that up-regulated DEPs predominantly enriched pathways including focal adhesion, cell-adhesion molecules, ECM-receptor interaction, and the PI3K-Akt signaling pathway. Meanwhile, down-regulated DEPs were mainly enriched in metabolic pathways, glycerolipid metabolism, arginine and proline metabolism, and tryptophan metabolism, among others (Figure 4c,d). The top ten KEGG-enriched signaling pathways for up-regulated and down-regulated DEPs are presented in Appendix A, respectively. These findings suggest that proteins abnormally expressed in SCNT bovine placentas may contribute to metabolic dysfunction, possibly leading to placental hypertrophy.

### 2.5. Metabolomic Analysis of SCNT and CON Bovine Placenta Tissues

Based on the results of the transcriptome and proteome analyses, we found that the genes and proteins abnormally expressed in the SCNT bovine placenta are mainly involved in regulating metabolism-related biological functions. To further investigate these findings, we performed a metabolomic analysis of the SCNT and CON placentas using a liquid chromatography-tandem mass spectrometry (LC-MS/MS) detection platform and performed multivariate statistical analysis to identify the metabolites. The PCA results demonstrated that the metabolites of the SCNT and CON placentas could be well differentiated, with the PC1 and the PC2 accounting for 70.50% and 10.90% of the total variance, respectively (Figure 5a). As illustrated in Figure 5b, partial least squares discriminant analysis (PLS-DA) model validation confirmed that, with decreasing component retention, both R2 and Q2 values decreased, indicating an upward trend in the regression line. This indicates that the model validation was successful, and the model was not overfitted. Together, these findings signify substantial differences in the metabolite profiles between SCNT and CON bovine placental tissues.

We screened the DEMs with a *p*-value < 0.05 and VIP-PLS-DA > 1 and multiple difference of up/down regulation = 1. A total of 270 differential expression metabolites (DEMs) were selected, of which 214 metabolites were up-regulated and 56 were down-regulated in SCNT bovine placenta (Figure 5c,d). The top 10 most DEMs are listed in Table 4. Hormone metabolites including 13,14-dihydro-15-keto-PGA2 and Prostaglandin F3a were significantly enriched, suggesting abnormal hormone synthesis and secretion functions in the placenta of SCNT bovine.

We performed KEGG pathway enrichment analysis on the screened DEMs to explore their biological functions. This analysis revealed significant enrichment in metabolically related pathways, including sphingolipid, glutathione, pyrimidine, glycerophospholipid, and phenylalanine metabolism (Figure 5e). The top 10 metabolic pathways are shown in Table 5. Specifically, sphingolipid metabolism and glycerophospholipid metabolism are involved in cell proliferation and cell differentiation, glutathione metabolism is involved in immune regulation, and phenylalanine metabolism is involved in the regulation of glucose and fat metabolism. These results suggest that abnormalities in placental cell proliferation, energy metabolism, and immune function, as revealed through the enrichment in these pathways, may impede the normal function of the SCNT bovine placenta.

### 2.6. Correlation Analysis of Transcriptomics and Metabolomics Data

To investigate the association between abnormally expressed metabolites and genes in SCNT bovine placental tissues more thoroughly, we conducted a combined transcriptomics and metabolomics analysis (Figure 6a). Venn diagrams revealed common pathways affected in both DEMs and DEGs identified in SCNT compared to CON placental tissues, specifically in choline metabolism and the biosynthesis of unsaturated fatty acids (Figure 6b, Table 6). Additionally, we tested the correlation between DEMs and DEGs based on their differential enrichment in the transcriptome and metabolome. Pearson correlation analysis was conducted on the top 10 metabolites and top 20 genes, resulting in an interaction network diagram for the top 40 genes and metabolites. The analysis identified 9 genes that were significantly positively or negatively associated with one or more of the top 10 metabolites (Figure 6c,d). Genes enriched through combined transcriptome and metabolome analyses, including phospholipase A2, group Ⅳ B (*PLA2G4B*), phosphatidylinositol-3-kinase catalytic subunit alpha (*PIK3CA*), son of sevenless-1 *(SOS1*), and solute carrier family 22 member 3 (*SLC22A3*), have been shown to play key roles in mammalian placental development.

## 3. Discussion

The placenta, a critical organ for gas exchange and nutrient exchange, serving as an immune barrier between the fetus and mother, is crucial for proper development throughout pregnancy. However, cloned animals often exhibit phenomena such as placental enlargement and abnormalities in trophoblast cells during pregnancy. In our previous study, we observed a high rate of abortion in SCNT cattle, accompanied by a significant increase in placental tissue weight and volume during late pregnancy [22]. Similarly, Chavatte et al. observed placental hypertrophy and fetal overgrowth in cloned cattle [23]. Stice et al. suggested that placental dysplasia in SCNT bovine, characterized by an increase in total placental weight and a decrease in placental vasculature, among other abnormalities, were primary causes of abortion [24]. In this study, transcriptomic, proteomic, and metabolomic analyses identified significant changes in gene expression, protein levels, and metabolites in SCNT bovine placental tissues compared to CON. Currently, some progress has been made in the study of placental abnormalities in SCNT animals through various omics techniques. Bang et al. detected differentially expressed proteins in the placentas of SCNT and CON cats using proteomics methods, finding that the abnormally expressed proteins in the placenta of SCNT cats were mainly associated with oxidative damage, aging, and apoptosis. This potentially leads to impaired placental development and dysfunction in the early gestation period and possibly induces miscarriage, resulting in the failure of the fetus to maintain pregnancy to term [25]. Kim et al. conducted a proteomic analysis of SCNT bovine placenta and found an abnormal expression of proteins involved in hormone production, fetal development, and the maintenance of maternal-fetal balance during pregnancy [26]. Proteomic analysis and characterization of SCNT pig placenta revealed that differential proteins may induce the apoptosis of placental trophoblast cells through mitochondria-mediated mechanisms, leading to placental chorion hypoplasia [27]. Gao et al. sequenced mRNA, lncRNA, and miRNA from cotyledonary tissues of SCNT bovine placentas in the late gestation period via transcriptome analysis, finding that DEGs were significantly enriched in functions related to urea and ion transmembrane translocation, and maternal-fetal interaction was impaired in SCNT bovine placenta [22]. Su et al. discovered, through Solexa sequencing, that differentially expressed miRNA were mainly related to the immune function of SCNT bovine placenta, where the abnormal expression of miR-136 may lead to placental defects, resulting in the fetal death of late SCNT fetus and/or neonatal SCNT fetuses [28]. Okae et al. analyzed expressions of placental intermediate genes in SCNT mice using transcriptome and found that allelic imprinting deletion was present in grb-2-associated binder 1 (Gab1), scm-like with four mbt domains 2 (Sfmbt2), and solute carrier family 38 member 4 (Slc38a4), which may contribute to the enlarged placenta in SCNT mice [29].

During pregnancy, the placenta functions not only as an immune organ but also as an active endocrine organ capable of secreting a variety of active substances essential for the normal growth and development of both the placenta and the fetus, such as progesterone, estrogen, placental prolactin, and pregnancy-related glycoproteins [30]. In this study, we discovered that DEGs are significantly enriched in the signaling pathways of growth hormone synthesis, secretion, and action. Metabolomic analysis revealed an abnormal expression of hormone metabolites such as 13,14-dihydro-15-keto-PGA2 and Prostaglandin F3a in SCNT bovine placenta, indicating interference with growth hormone synthesis and secretion. Consistent with previous reports, growth-related hormones such as P4, PAGs, and estradiol are abnormally secreted in SCNT bovine placentae during pregnancy [31,32,33]. Additionally, our study found that DEGs and DEPs were primarily associated with metabolic function, and KEGG enrichment analyses of DEGs, DEPs, and DEMs are significantly associated with metabolism-related signaling pathways, including glycolysis/gluconeogensis, metabolic pathways, glycerolipid metabolism, and arginine and proline metabolism. Glucose, a crucial metabolic substrate, provides energy for life activities through glycolysis or gluconeogenesis [34]. Glycerides function as energy-storage molecules, playing a role in energy metabolism [35]. Arginine and proline, parts of the glutamate family of amino acids, with glutamate serving not only as the precursor metabolite to proline but also as the final product of the degradation of proline and arginine [36,37]. Arginine can be converted into ornithine and urea through the urea cycle, while proline can be synthesized from glutamate and eventually converted into pyruvate to participate in the TCA cycle [38]. These findings indicate that various energy-metabolic processes, including glycolysis, the TCA cycle, and oxidative phosphorylation, are abnormal in SCNT bovine placenta. These abnormalities further affect the activity and function of SCNT placental cells, hinder the normal operation of SCNT bovine placental function, and lead to placental hypertrophy, resulting in pregnancy failure and an increased morbidity and mortality of offspring in SCNT cattle.

Moreover, we found that the TKDP1, TKDP4, TIMP4 genes, and CSF1 proteins were aberrantly expressed in SCNT bovine placenta. TKDP1 and TKDP4 belong to a class of multigene families with specific high expression in ruminant hoofed animal placentas, acting as key transcription factors in the process of placental trophoblastic differentiation [39]. TIMP4, serving as a tissue metalloproteinase inhibitor, can inhibit trophoblastic invasion and migration in the presence of long non-coding RNA, which is associated with the pathogenesis of preeclampsia [40]. The CSF1 protein is pivotal in regulating placental trophoblast cell proliferation and development, contributing to conditions such as preeclampsia [41,42]. Meanwhile, we found an abnormal expression of MKI67 and IMPDH1 genes involved in cell growth and proliferation, as well as the proteins ITGA2, ITGB4, and GARRE1 in SCNT bovine placental tissues. This study also revealed differentially expressed genes and proteins in SCNT bovine placenta associated with protein digestion and absorption, indicating abnormal protein synthesis and expression in SCNT bovine placenta.

Previous reports have indicated an abnormal expression of angiogenesis-related proteins, pregnancy-associated proteins, SOD, tropomyosin, and PAI1 in SCNT bovine placentas [43,44]. The combined analysis of DEGs and DEMs further identified key genes like PLA2G4B, PIK3CA, SOS1, and SLC22A3, which play significant roles in mammalian placental development. PLA2G4B, a phospholipase involved in the delivery process, regulates gene expression primarily through interactions with miRNAs, and it impacts the human placenta’s delivery process [45]. PIK3CA, a regulatory gene involved in growth and metabolism, has shown impaired placental growth and structure in mice with PIK3CA knockout but an enhanced ability to supply glucose and amino acids [46]. The SOS1 gene is essential for fetal development, with its deletion leading to fetal death in the second trimester of gestation. Qian et al. have shown that placental development in mice targeted with a deletion of SOS1 is impaired, resulting in fetal death in the second trimester [47]. Genomic imprinted genes play an important role in mammalian gamete production, as well as embryonic and placental growth and development [48]. SLC22A3, an imprinted gene in the IGF2R imprinting region, has been shown to be conserved in mammalian placenta [49]. Aberrant expressions of imprinted genes such as Phlda2, Grb10, Mest, and Igf2 were observed in the placentas of cloned mice and cloned cattle [50,51,52]. The abnormal expression of imprinted genes leads to dysfunction in the proliferation and differentiation of placental trophoblastic cells, defects in placental growth and development, and a reduced nutrient-transport capacity, which affect pregnancy in SCNT animals. These results further suggest that the abnormal expression of genes in the placenta of SCNT cattle affects the morphology and function of the placenta, leading to the hypertrophy of the placenta of SCNT cattle, and it may also be an important cause of the low pregnancy success rate of SCNT cattle.

## 4. Materials and Methods

### 4.1. Ethics Approval and Consent to Participate

All experimental procedures in this study were consistent with the National Research Council Guide for the Care and Use of Laboratory Animals. All protocols were approved by the Institutional Animal Care and Use Committee at Inner Mongolia University (approval number: IMU-CATTLE-2022-063).

### 4.2. SCNT Animals and Sample Collection

Collect ovaries from a slaughterhouse, and aspirate follicular fluid from antral follicles with a diameter of 3–8 mm using a syringe. Under a stereomicroscope, pick the cumulus-oocyte complexes, and wash them in a washing medium. Then, mature the oocytes in an in vitro maturation medium at 38.5 °C with 5% CO_2_ in a four-well culture plate for 18–22 h. After maturation, remove the cumulus cells from the oocytes by rotating them in an M199 medium (12340-030, Gibco, Waltham, MA, USA) containing 0.1% hyaluronidase (935166, Sigma, St. Louis, MO, USA). Select 20–30 oocytes, and place them into a 30-µL nuclear-transfer solution droplet (M19390L415L9 + 1% FBS + 5 µg/mL CB) covered with mineral oil. Remove the first polar body and adjacent cytoplasm containing the oocyte chromosomes. Insert a single donor cell into the perivitelline space of the oocyte. In this study, the SCNT somatic cells were fibroblasts from the ear tip of a 2-year-old, high-quality bull. Place the cytoplast-cell complexes into an embryo development medium. After 30 min of recovery, transfer the reconstructed embryos into a 1-mm fusion chamber, and apply electrical pulses of 20 µs per pulse at 125 V/mm for fusion. Subsequently, culture the embryos in development medium at 38.5 °C with 5% CO_2_ and saturated humidity for 30 min. Under conditions of 5% CO_2_ and saturated humidity, first, incubate the embryos with 5 µmol/L of ionomycin (13909, Sigma, USA) for 5 min, and then culture with 10 µg/mL of cytochalasin B (C6762, Sigma, USA) at 38.5 °C with 5% CO_2_ and saturated humidity for 5 h. After activation, culture the embryos in embryo development medium (SOF + 5% FBS) for 48 h, and then co-cultivate the cleaved reconstructed embryos with cumulus cells until they reach the blastocyst stage before embryo transfer into the uterus of recipient cows. CON cows undergo artificial insemination simultaneously. Both groups of cows are fed the same diet in the same environment. At parturition, collect placental tissues from CON and SCNT cows, cut them into 1–2 cm pieces, and immediately freeze them in liquid nitrogen. Store the samples at −80 °C to preserve RNA, proteins, and metabolites.

### 4.3. Transcriptomic Sequencing and Analysis

Total RNA was extracted from the placental tissues of CON and SCNT cattle using Trizol reagent (Invitrogen, Shanghai, China), following the manufacturer’s instructions. The total RNA concentration and purity were measured with a Nanodrop2000, and RNA integrity was assessed via agarose-gel electrophoresis. The cDNA libraries were prepared following the Illumina^®^ Stranded mRNA Prep, Ligation (San Diego, CA, USA). A total of 6 libraries were constructed with 3 biological replicates per group. The cDNA synthesis involved end repair, phosphorylation, and A-tailing, as per Illumina’s protocol for library construction. Libraries were size-selected on 2% Low Range Ultra Agarose to isolate cDNA target for 300 bp, followed by amplification using Phusion DNA polymerase (NEB) over 15 PCR cycles. After being quantified via Qubit 4.0, the paired-end RNA-seq sequencing libraries were sequenced on a NovaSeq 6000 sequencer (2 × 150 bp read length). Reads were aligned to the bovine reference genome using Bowtie (ARS-UCD1.2). To identify DEGs between two different samples, the expression level of each transcript was calculated according to the TPM method. Differential expression analysis was performed using the DEG-Seq software (Version 1.38.0). DEGs with a *p*-value < 0.05 and |Log2 FC| ≥ 1 were considered significantly differentially expressed genes. Functional enrichment analyses of the GO and KEGG pathways were conducted using Goatools (Version 0.6.5) and Python’s scipy software (Version 1.4.1), respectively.

### 4.4. Protein Sequencing and Analysis

Total proteins were extracted from CON and SCNT bovine placenta tissues. After protein extraction, a total of six libraries were constructed from three biological replicates for each group. The protein concentration was determined using the BCA (V900933, Sigma, USA) method. Digest the proteins into peptides using trypsin. The peptides were labeled with TMT reagents, using different TMT labels for different samples. The labeled peptides from different samples were mixed together for simultaneous analysis. The mixed peptides were separated using high-performance liquid chromatography. Peptide mass spectrometry analysis was performed using a mass spectrometer. TMT labels enable a quantitative comparison of peptides in different samples by detecting labeled ions in the MS/MS analysis. All proteins and protein sequences identified were compared against databases including EggNOG, GO, KEGG, NR, Pfam, String, and Uniprot. Based on expression quantities, differential proteins between the groups were analyzed using Student’s *t*-test, with the threshold for differential protein screening set to |log2FC| ≥ 0.585 and a *p*-value < 0.05. DEPs underwent GO functional enrichment and KEGG pathway enrichment analysis using Goatools 0.6.5 and the scipy library in Python, respectively.

### 4.5. Metabolomics Analysis

Total metabolites were extracted from CON and SCNT bovine placental tissues using three biological replicates per group to construct a total of six libraries. For each extraction, a 50-mg solid sample was accurately weighed, and the metabolites were extracted using a 400 µL methanol-water (4:1, *v*/*v*) solution. The mixture was allowed to settle at −20 °C and treated with a high-throughput tissue crusher, Wonbio-96c (Shanghai, China), at 50 Hz for 6 min, which was then followed by vortex for 30 s and ultrasound at 40 kHz for 30 min at 5 °C. The samples were placed at −20 °C for 30 min to precipitate proteins. After centrifugation at 13,000× *g* and 4 °C for 15 min, the supernatant was carefully transferred to sample vials for LC-MS/MS (LC-Bio, Hangzhou, China) analysis. Quality-control (QC) samples were prepared by mixing equal volumes of all metabolite samples, and a QC sample was inserted every three samples to detect the reproducibility of the entire analysis process. Metabolites were separated using a BEH C18 chromatographic column (100 mm × 2.1 mm i.d., 1.8 µm) before entering mass spectrometry detection. The separated metabolites were then analyzed via mass spectrometry in positive and negative ion modes. After the machine operation was completed, Progenesis QI (Waters Corporation Milford, MA, USA) software (Version 2.0) was used to perform baseline filtering, peak identification, integration, retention-time correction, and peak alignment on the raw data, and the resulting data matrix was normalized and subjected to log10 logarithmic processing. The MS and MSMS mass spectrometry information was matched with the public metabolite databases HMDB (http://www.hmdb.ca/, accessed on 1 September 2020) and Metlin (https://metlin.scripps.edu/landing_page.php?pgcontent=mainPage, accessed on 1 September 2020) to annotate the metabolites. The R software package ropls (Version 1.6.2) was used for PCA and OPLS-DA analysis, and 7-fold cross-validation was used to evaluate the stability of the model. Differential metabolites were screened using univariate statistical analysis (*t*-test) combined with OPLS-DA/PLS-DA and FC, with the screening criteria at *p* < 0.05 and VIP > 1. Pathway enrichment analysis of the identified DEMs was conducted using KEGG.

### 4.6. Combined Metabolome and Transcriptome Analysis

A joint analysis of the metabolome and transcriptome was performed to identify overlapping KEGG pathways, which were visualized using a Venn diagram tool. In order to study the correlation between genes and metabolites, Pearson correlation analysis was utilized. Based on the results, correlation heatmaps for samples and network diagrams were created to illustrate the interactions.

### 4.7. Data Availability Statement

The datasets produced and/or analyzed during the current study are available from the corresponding author upon reasonable request. RNA-seq data were deposited in the GSA under the accession number CRA014398.

## 5. Conclusions

In this study, we analyzed transcriptomic, proteomic, and metabolomic data from CON and SCNT bovine placental tissues. We found that DEGs, DEPs, and DEMs were enriched in functional pathways including growth hormone synthesis, secretion, and action, protein digestion and absorption, glycolysis/gluconeogensis, metabolic pathways, glycerolipid metabolism, and arginine and proline metabolism. Furthermore, key genes and proteins involved in the regulation of placental trophoblast development and cell proliferation, as well as those related to placental morphology, structure, growth, and development, were also abnormally expressed in SCNT bovine placenta. In summary, our results suggest that trophoblast cell proliferation, development, growth hormone synthesis and secretion, protein synthesis, and energy metabolism are abnormal in SCNT bovine placenta. These abnormalities may impact the proper placental morphology, structure, and function of SCNT cattle, potentially leading to placental hypertrophy and, ultimately, pregnancy failure. This study provides valuable data for further investigation into the mechanisms of SCNT bovine placental hypertrophy. However, further validation and research are essential to elucidate the specific biological mechanisms driving this phenomenon during gestation.

## Figures and Tables

**Figure 1 ijms-25-09388-f001:**
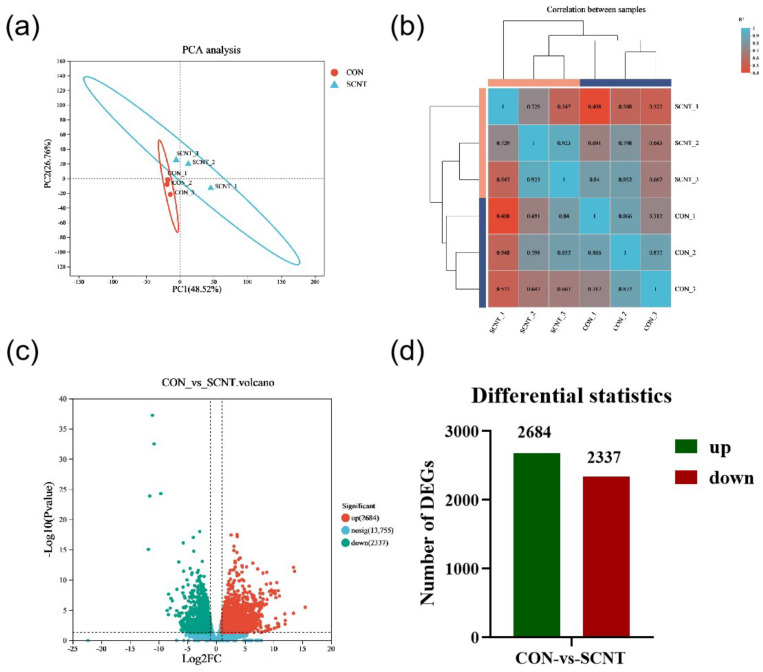
Transcriptome analysis of CON and SCNT bovine placental tissues. (**a**) PCA score plot of transcriptome of CON and SCNT bovine placental tissues. (**b**) Pearson correlation between CON and SCNT bovine placental tissue samples. (**c**) Volcano plot of DEGs in CON and SCNT bovine placental tissues (*n* = 3). (**d**) Number of DEGs up-regulated and down-regulated.

**Figure 2 ijms-25-09388-f002:**
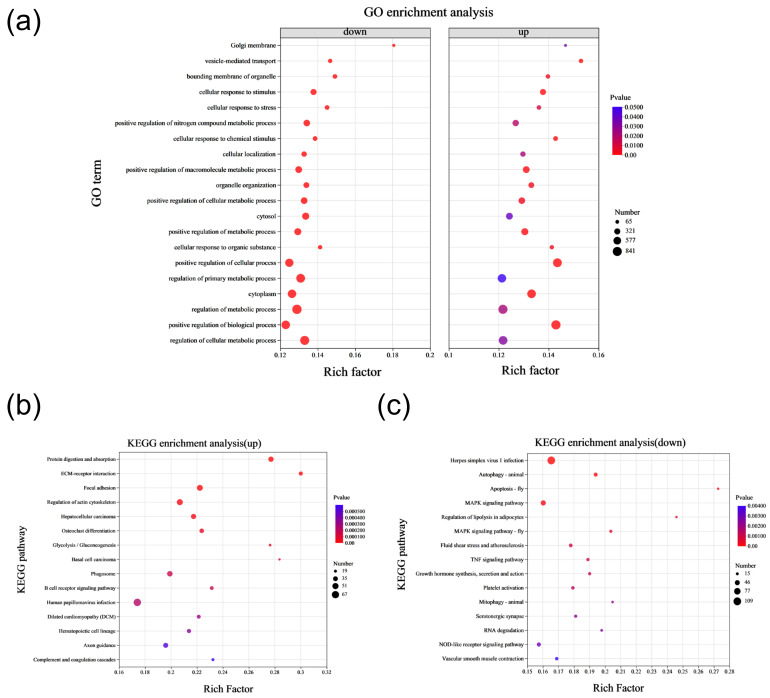
Enrichment analysis of DEGs in CON and SCNT bovine placental tissues. (**a**) GO enrichment analysis of up-regulated and down-regulated DEGs in the transcriptome of CON and SCNT bovine placentas. (**b**) KEGG enrichment analysis of up-regulated DEGs. (**c**) KEGG enrichment analysis of down-regulated DEGs.

**Figure 3 ijms-25-09388-f003:**
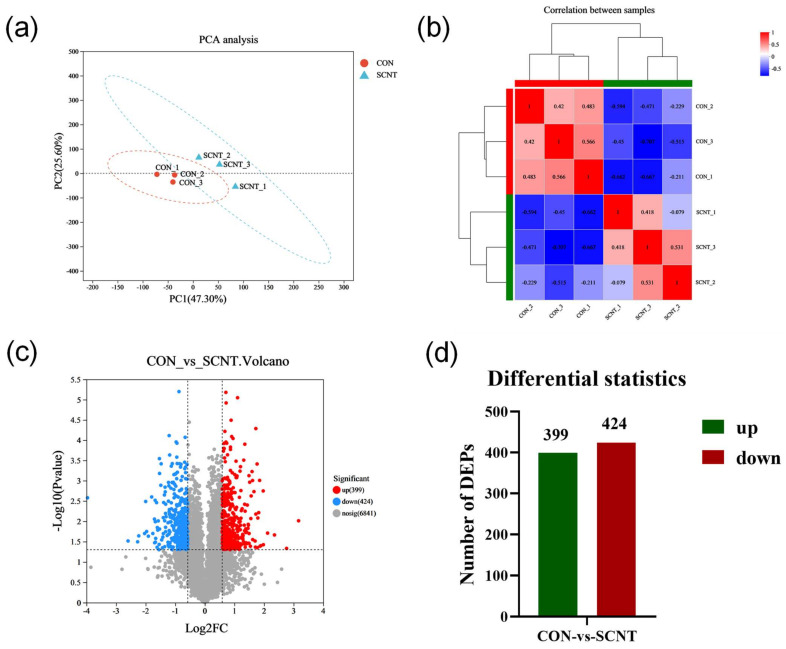
Proteomic analysis of CON and SCNT cattle placenta. (**a**) Plot of protein PCA scores of CON and SCNT cattle placental tissues. (**b**) Heat map of sample correlation of proteomics of CON and SCNT cattle placental tissues. (**c**) Volcano plot demonstrating differentially expressed proteins in CON and SCNT bovine placental tissues. (**d**) Number of DEPs with up-regulated expression and down-regulated expression.

**Figure 4 ijms-25-09388-f004:**
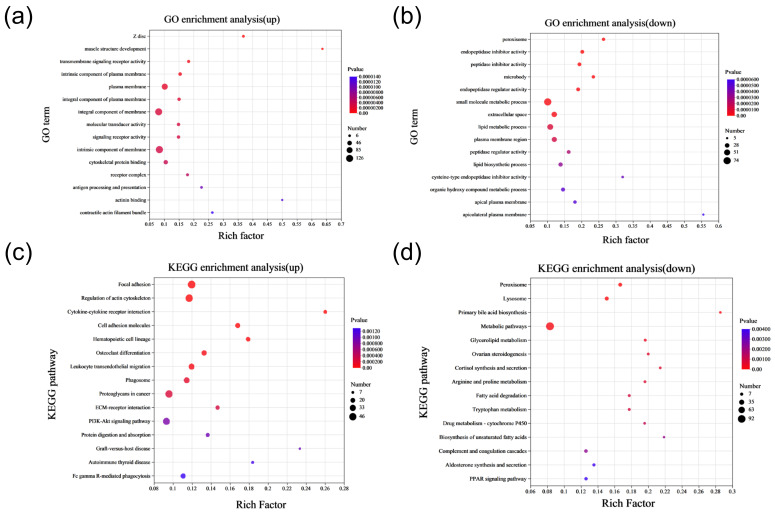
Enrichment analysis of DEPs in placental tissues of CON and SCNT bovine. (**a**) GO enrichment analysis of up-regulated DEPs. (**b**) GO enrichment analysis of down-regulated DEPs. (**c**) KEGG pathway enrichment analysis of up-regulated DEPs. (**d**) KEGG pathway enrichment analysis of down-regulated DEPs.

**Figure 5 ijms-25-09388-f005:**
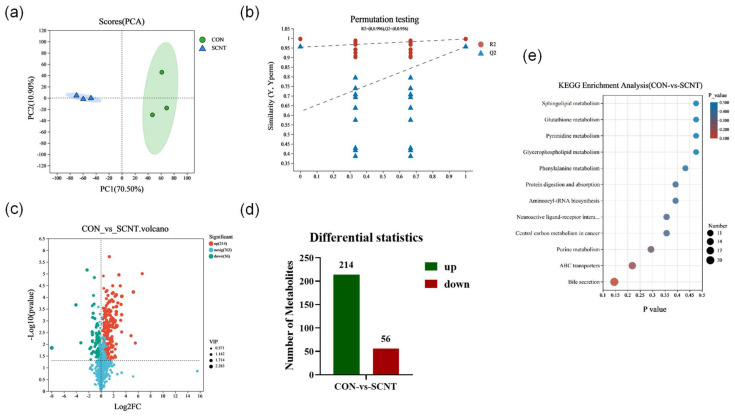
Metabolomics analysis of CON and SCNT bovine placental tissues. (**a**) PCA analysis plot of metabolomics of CON and SCNT bovine placental tissues. (**b**) Validation of PLS-DA substitution in metabolomes of CON and SCNT bovine placental tissues. (**c**) Volcano plot demonstrating DEMs in placental tissues of CON and SCNT bovine. (**d**) The number of up-regulated and down-regulated DEMs in placental tissues of CON and SCNT bovine. (**e**) Enrichment analysis of KEGG pathway for DEMs in placental tissues of CON and SCNT bovine.

**Figure 6 ijms-25-09388-f006:**
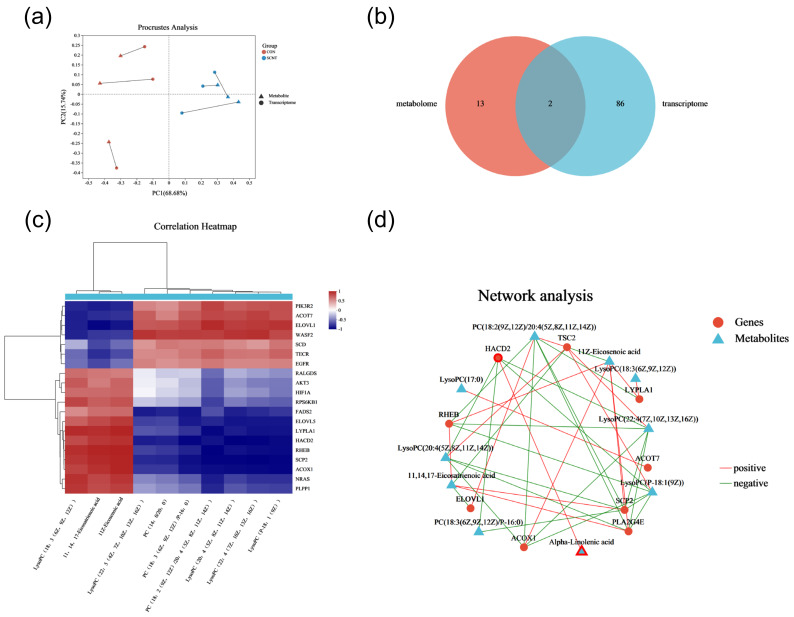
Combined transcriptomics and metabolomics analysis of CON and SCNT bovine placental tissues. (**a**) Schematic diagram of transcriptome and metabolome of CON and SCNT bovine placenta tissues. (**b**) Venn diagram of KEGG pathway co-enriched between transcriptomes and metabolomes. (**c**) Heatmap of correlation between DEGs and DEMs. (**d**) Network diagram of correlation of DEGs and DEMs.

**Table 1 ijms-25-09388-t001:** Summary of the transcriptome sequencing data.

Sample	Raw Reads	Raw Bases	Clean Reads	Clean Bases	Error Rate (%)	Q20 (%)	Q30 (%)	GC Content (%)
CON_1	47,653,834	7,195,728,934	47,052,430	6,941,489,461	0.0246	98.11	94.58	51.13
CON_2	49,275,204	7,440,555,804	48,669,080	7,107,312,975	0.0244	98.19	94.81	52.5
CON_3	41,920,862	6,330,050,162	41,293,700	6,084,502,265	0.0247	98.05	94.49	52.49
SCNT_1	44,027,118	6,648,094,818	43,098,966	6,291,349,718	0.0243	98.21	94.95	57.15
SCNT_2	42,114,108	6,359,230,308	41,502,758	6,139,152,712	0.0243	98.25	94.97	55.06
SCNT_3	49,399,126	7,459,268,026	48,727,536	7,177,370,735	0.0248	98.05	94.45	53.86

**Table 2 ijms-25-09388-t002:** Top 10 DEGs.

Gene Name	*p*-Value	Log2 FC	Regulation
ENSBTAG00000054274	5.93 × 10^−38^	−11.10993504	Down
TKDP4	3.15 × 10^−33^	−10.8104713	Down
CYP4B1	5.47 × 10^−25^	−9.657055193	Down
TKDP1	1.35 × 10^−24^	−11.57468493	Down
TIMP4	1.01 × 10^−18^	−2.87067796	Down
MKI67	3.15 × 10^−18^	3.621668856	Up
IMPDH1	3.61 × 10^−18^	2.597485953	Up
EPB41L1	6.47 × 10^−18^	3.666560722	Up
ENSBTAG00000037799	9.37 × 10^−18^	−3.982927754	Down
ENSBTAG00000053827	7.48 × 10^−17^	−5.731321654	Down

**Table 3 ijms-25-09388-t003:** Top 10 DEPs in placental tissue of SCNT and CON bovine.

Protein Name	*p*-Value	Log2 FC	Regulation
ITGB4	6.00 × 10^−6^	−0.875523	Down
ABCC1	7.00 × 10^−6^	0.709874	Up
PDLIM7	9.00 × 10^−6^	1.105236	Up
GARRE1	1.20 × 10^−5^	0.718408	Up
MICAL1	3.20 × 10^−5^	0.886415	Up
CSF1	5.20 × 10^−5^	1.722665	Up
OLFML3	6.10 × 10^−5^	0.676249	Up
OR10AG63	7.70 × 10^−5^	−1.206875	Down
ITGA2	8.10 × 10^−5^	0.907248	Up
NAD3	8.50 × 10^−5^	−0.665457	Down

**Table 4 ijms-25-09388-t004:** The top ten differential metabolites of CON and SCNT bovine placenta.

Metabolite	*p*-Value	FC	VIP	Regulate
Hoduloside VII	1.88 × 10^−6^	2.5188	1.5456	Up
PC(18:2(9Z,12Z)/20:4(5Z,8Z,11Z,14Z))	6.90 × 10^−6^	0.2083	1.691	Down
Nomilinic acid	9.91 × 10^−6^	101.0917	1.5009	Up
Yucalexin P21	1.11 × 10^−5^	7.3884	1.7241	Up
13,14-dihydro-15-keto-PGA2	1.25 × 10^−5^	1.3036	1.0464	Up
1-hexadecyl-glycero-3-phosphate	1.45 × 10^−5^	0.4739	1.3535	Down
(3S,7E,9R)-4,7-Megastigmadiene-3,9-diol 9-[apiosyl-(1- > 6)-glucoside]	3.26 × 10^−5^	3.6881	1.5094	Up
2-Methyl-3-(2-pentenyl)-2-cyclopenten-1-one	3.27 × 10^−5^	8.5901	1.5401	Up
Prostaglandin F3a	4.22 × 10^−5^	1.3644	1.1154	Up
11Z-Eicosenoic acid	5.18 × 10^−5^	1.9694	1.4041	Up

**Table 5 ijms-25-09388-t005:** Top 10 metabolic KEGG pathways.

Pathway ID	Pathway Name	*p*-Value
map04976	Bile secretion	0.1458
map02010	ABC transporters	0.2184
map00230	Purine metabolism	0.2937
map05230	Central carbon metabolism in cancer	0.3569
map04080	Neuroactive ligand-receptor interaction	0.3569
map00970	Aminoacyl-tRNA biosynthesis	0.393
map04974	Protein digestion and absorption	0.393
map00360	Phenylalanine metabolism	0.4326
map00240	Pyrimidine metabolism	0.4758
map00480	Glutathione metabolism	0.4758

**Table 6 ijms-25-09388-t006:** Shared KEGG pathway between the transcriptome and metabolome.

KEGG Pathway	KEGG ID	Metabolites	Genes
Choline metabolism in cancer	bta05231	PC(18:2(9Z, 12Z)/20:4(5Z, 8Z, 11Z, 14Z)), PC(14:0/20:0), PC(18:3(6Z, 9Z, 12Z)/P-16:0), LysoPC(20:4(5Z, 8Z, 11Z, 14Z)), LysoPC(18:3(6Z, 9Z, 12Z)), LysoPC(22:4(7Z, 10Z, 13Z, 16Z)), LysoPC(22:5(4Z, 7Z, 10Z, 13Z, 16Z)), LysoPC(P-18:1(9Z)), LysoPC(17:0)	RPS6KB1, MAPK8, RALGDS, PLA2G4B, NRAS, PIK3CA, PIK3R2, AKT3, SLC22A2, HIF1A, SLC44A4, EGFR, PIK3CD, PDGFRA, PLA2G4E, RHEB, GPCPD1, LYPLA1, SOS2, TSC2, DGKG, SLC44A3, PDGFD, SLC22A3, DGKI, PLPP1, KRAS, WASF2, WAS, PIP5K1B
Biosynthesis of unsaturated fatty acids	bta01040	11Z-Eicosenoic acid, Alpha-Linolenic acid, 11,14,17-Eicosatrienoic acid	ELOVL1, HACD2, HACD1, SCP2, ACOT7, ACOX1, SCD, ELOVL5, ELOVL4, FADS2, ENSBTAG00000054697, TECR, FADS1

## Data Availability

The data presented in this study are available upon request from the corresponding author.

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
