# Peer review of "Integrating Transcriptomics, Proteomics, and Metabolomics to Investigate the Mechanism of Fetal Placental Overgrowth in Somatic Cell Nuclear Transfer Cattle"

_ijms, 2024, doi:10.3390/ijms25179388_

Round 1

Reviewer 1 Report

Comments and Suggestions for Authors

The manuscript presents a comprehensive study integrating multiple omics approaches to investigate placental overgrowth in SCNT cattle. However, several methodological details need further elaboration to ensure the study's reproducibility and clarity. This will not only improve the paper's readability but also strengthen its scientific rigor.

L3: Could authors not to use abbreviation in the title.

L17: Give full name for an abbreviation first appears in the manuscript.

L22: please use full name where this is first mentioned in the manuscript

L236: Can authors give more details on how these metabolites identified with confidence?

L345-L346: Can authors add more details about the animals and sample collection including more details about the SCNT and how stored and if any QC done on that. This is a major part of the study and need to be detailed

L372: Can authors add more details for the TMT mass spectrometry experiment

L385-386: Can authors explain how methanol with water can form lower organic layer? It is known that methanol and water form one layer?

L386: Can authors add more details for the metabolomics method

L403-408: Can authors explain these lines and how are they relevant?

Author Response

请看附件

Reviewer 2 Report

Comments and Suggestions for Authors

The study addresses a relevant and important topic in animal cloning, particularly focusing on the abnormalities in placental development in SCNT (Somatic Cell Nuclear Transfer) cattle. The integration of multiple omics analyses (transcriptomics, proteomics, and metabolomics) is noted as a strength of the study. The research provides valuable insights into the mechanisms of placental overgrowth in SCNT cattle, which is a significant issue in the field of animal cloning and reproductive biotechnology. The methods are generally well-described, and the multi-omics approach is appropriately applied. However, some additional details or clarifications might be necessary to ensure reproducibility. The data appears to be robust and supports the conclusions drawn in the paper. The study uses appropriate statistical methods to analyze the data. The manuscript is well-structured, with clear figures and tables that support the text. However, some minor language corrections may be needed to improve readability.

The abstract effectively summarizes the study's objectives, methods, results, and conclusions. However, it could be more concise. The introduction provides a solid background on SCNT technology and the significance of placental abnormalities in cloned animals. It could benefit from more discussion on the potential implications of the study's findings. The methods section is comprehensive, covering the details of sample collection, omics analyses, and statistical methods. Some minor clarifications, such as on the selection criteria for DEGs (Differentially Expressed Genes) and DEPs (Differentially Expressed Proteins), could enhance the clarity. The results are well-presented with appropriate use of figures and tables. The study finds significant differences in gene, protein, and metabolite expression between SCNT and control cattle placentas, supporting the hypothesis of abnormal placental development in SCNT cattle.

The discussion interprets the results in the context of previous studies and highlights the significance of the findings. The authors effectively link the observed omics changes to potential mechanisms of placental overgrowth. However, the discussion could be expanded to consider the broader implications of these findings for SCNT technology and animal cloning. The conclusion is consistent with the study's findings and summarizes the key points effectively.

Comments on the Quality of English Language

I did not detect any major issues with the English language, such as incomprehensible phrases or significant editing errors. The language used in the document is generally clear and precise, suitable for a scientific paper. However, there are some minor language and stylistic issues that could be improved for better readability. These include:

 Some sentences could be restructured for better flow and clarity. For instance, the use of passive voice is quite frequent, which is common in scientific writing, but in some cases, active voice could make the sentences clearer.

 In a few places, the text could be made more concise. For example, the abstract could be shortened slightly without losing essential details, improving its impact.

 There are a few instances of repetitive phrasing that could be revised to avoid redundancy and make the text more engaging.

While the use of technical terms is necessary in a scientific paper, ensuring that they are defined or explained, especially in the introduction, can help broader audiences understand the work better.

There might be a few punctuation errors, such as missing commas or periods, which should be corrected to maintain the formal tone of the paper.

Here are some suggestions that might contribute to a more concise text:

In the Abstract:

Original: "Our results indicate that abnormal protein synthesis, growth hormone function, and energy metabolism in SCNT bovine placental tissues lead to the occurrence of placental hypertrophy."

Improvement: "Our results suggest that abnormal protein synthesis, growth hormone function, and energy metabolism in SCNT bovine placental tissues contribute to placental hypertrophy."

Original: "These findings provide a valuable reference for an in-depth study of the mechanism of SCNT bovine placental abnormalities."

Improvement: "These findings offer valuable insights for further investigation into the mechanisms underlying SCNT bovine placental abnormalities."

In the Introduction:

-Original: "Somatic cell nuclear transfer (SCNT) is an asexual reproduction technique in which the nucleus of an animal somatic cell is transplanted into a denucleated oocyte to obtain a reconstructed embryo."

Improvement: "Somatic cell nuclear transfer (SCNT) is an asexual reproduction technique where the nucleus of a somatic cell is transplanted into a denucleated oocyte to create a reconstructed embryo."

Original: "However, few studies have been conducted on the mechanism of abnormal placental development in SCNT animals during pregnancy."

Improvement: "However, few studies have explored the mechanisms of abnormal placental development in SCNT animals during pregnancy."
